# Progressive Parameter Efficient Transfer Learning for Semantic Segmentation

**Nan Zhou**[1,2*], **Huiqun Wang**[1,2*], **Yaoyan Zheng**[1,2], **Di Huang**[1,2†]
[1]State Key Laboratory of Complex and Critical Software Environment, Beihang University, Beijing, China
[2]School of Computer Science and Engineering, Beihang University, Beijing, China
{zhounan0431,hqwangscse,yaoyanzheng,dhuang}@buaa.edu.cn

## Abstract

Parameter Efficient Transfer Learning (PETL) excels in downstream classification fine-tuning with minimal computational overhead, demonstrating its potential within the pre-train and fine-tune paradigm. However, recent PETL methods consistently struggle when fine-tuning for semantic segmentation tasks, limiting their broader applicability. In this paper, we identify that fine-tuning for semantic segmentation requires larger parameter adjustments due to shifts in semantic perception granularity. Current PETL approaches are unable to effectively accommodate these shifts, leading to significant performance degradation. To address this, we introduce ProPETL, a novel approach that incorporates an additional midstream adaptation to progressively align pre-trained models for segmentation tasks. Through this process, ProPETL achieves state-of-the-art performance on most segmentation benchmarks and, for the first time, surpasses full fine-tuning on the challenging COCO-Stuff10k dataset. Furthermore, ProPETL demonstrates strong generalization across various pre-trained models and scenarios, highlighting its effectiveness and versatility for broader adoption in segmentation tasks. Code is available at: https://github.com/weeknan/ProPETL.

## 1 Introduction

Parameter Efficient Transfer Learning (PETL) aims to leverage the representational knowledge from large-scale pre-trained models (e.g., MAE (He et al., 2022) and DINOv2 (Oquab et al., 2023)) for downstream tasks while minimizing computational costs. Recent studies (Jia et al., 2022; Chen et al., 2022; Hu et al., 2022) show that PETL can match or even surpass the performance of full fine-tuning for downstream classification tasks, achieving this with less than one percent of the parameters needing adjustment. This efficiency has motivated researchers to explore its application across various computer vision scenarios (Han et al., 2024).

However, existing PETL methods face performance bottlenecks when applied to semantic segmentation (Jia et al., 2022; Hu et al., 2022). Unlike classification tasks, semantic segmentation requires the model to develop fine-grained perceptual capability to predict semantic labels for each pixel. This shift in perceptual granularity complicates the fine-tuning process for segmentation. The discrepancy is further illustrated in Fig. 1(a). Despite their superior results in classification tasks, PETL methods consistently exhibit a performance gap compared to full fine-tuning across most segmentation benchmarks. Consequently, full fine-tuning remains the preferred approach in current segmentation efforts, limiting PETL's broader adoption.

Upon examining the fine-tuning process for classification and segmentation, we identify two key phenomena. First, transitioning from a classification pre-trained model to a downstream segmentation task requires extensive parameter adjustments during full fine-tuning, as indicated by the increased mean Euclidean distance in Fig. 1(b). Second, the limited tunable parameters in PETL hinder its ability to capture semantically-aware changes when fine-tuning directly for segmentation. As illustrated in Fig. 1(c), the PETL method, AdaptFormer (Chen et al., 2022), exhibits small adjust-

---

[*]Equal contribution
[†]Corresponding author.

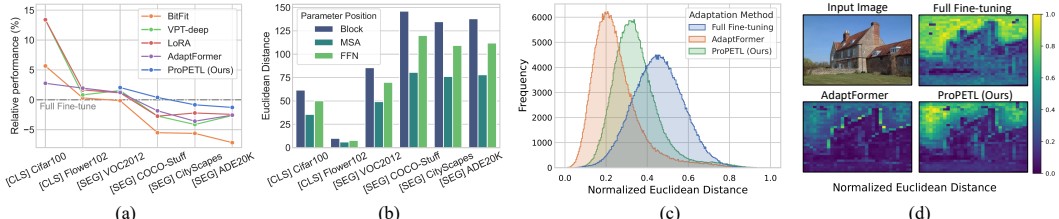

Figure 1: (a) Relative performance of PETL methods compared to full fine-tuning on image classification [CLS] and semantic segmentation [SEG] tasks. (b) Euclidean distance between the parameter vectors of the classification pre-trained model before and after full fine-tuning across different downstream tasks. The optimizer settings and the number of training iterations are aligned to minimize the influence of training conditions and data volume. (c)Histogram showing the normalized Euclidean distance between the feature map of the pre-trained model and the feature map after adaptation by different methods. (d) A visualization from ADE20k, illustrating the normalized Euclidean distance map after applying various adaptation methods.

ment magnitudes across most regions of the feature maps, unlike the uniformly distributed adjustments observed in full fine-tuning, resulting in a statistically right-skewed distribution. This trend is further evident in the samples visualized in Fig. 1(d). These observations indicate that fine-tuning for segmentation tasks demands both larger parameter adjustments and broader semantic changes to bridge the perceptual granularity gap between pre-trained models and downstream tasks. However, existing PETL methods struggle with these shifts, leading to suboptimal segmentation performance.

These insights motivate us to develop a solution that retains PETL's computational efficiency while adapting to the significant changes in semantic perceptual granularity during fine-tuning. Inspired by previous works (Hsu et al., 2020; Dong et al., 2024b) that improve domain generalization through a progressive paradigm, we introduce an additional phase, midstream adaptation, into the PETL framework to progressively address semantic granularity differences. Achieving this involves two major challenges. First, developing a strategy to bridge the semantic perception gap remains underexplored. Unlike prior adaptations that address data domain changes (Dong et al., 2024b), the progressive adaptation for segmentation must effectively bridge the differences in semantic perceptual granularity between upstream and downstream tasks, requiring detailed analysis and modeling. Second, it remains uncertain which level of supervision and specific perceptual granularity for intermediate tasks would most benefit progressive fine-tuning for segmentation. Identifying the most effective intermediate task for enhancing segmentation perception remains an open question.

To address these challenges, we begin with analyzing two candidate progressive adaptation strategies: Generalized Parametric Adaptation (GPA) and Decoupled Structured Adaptation (DSA). We then conduct a comprehensive investigation into the impact of different intermediate tasks and empirically determine the optimal selection. By integrating both approaches, we propose ProPETL, a novel progressive PETL framework for segmentation adaptation. By effectively bridging the perception gap between pre-trained representational knowledge and downstream segmentation tasks, ProPETL significantly enhances performance while maintaining computational efficiency. Compared to its counterparts, ProPETL substantially narrows the performance gap with full fine-tuning on most benchmarks and, for the first time, surpasses full fine-tuning on the challenging COCO-Stuff10k dataset, demonstrating its effectiveness.

Our contributions are summarized as follows:

**1)** We introduce ProPETL, an innovative framework for progressively adapting pre-trained models to downstream semantic segmentation tasks through intermediate tasks.

**2)** We conduct an in-depth analysis and comprehensive comparison of candidate progressive adaptation strategies and intermediate tasks to facilitate the continuous migration of representational knowledge from pre-trained models to segmentation tasks.

**3)** We achieve state-of-the-art segmentation fine-tuning performance across diverse benchmarks and demonstrate robust generalizability across various pre-trained models and segmentation tasks.

## 2 RELATED WORK

### 2.1 PARAMETER EFFICIENT TRANSFER LEARNING

PETL aims to effectively fine-tune pre-trained models for downstream tasks with minimal parameter updates. Recent approaches can be categorized into three main types: partial tuning, prompt-based, and adapter-based methods. Partial tuning methods strategically select a subset of the pre-trained model for fine-tuning. For example, BitFit (Zaken et al., 2022) updates only the bias terms of the network while freezing other parameters, and SPT (He et al., 2023) calculates a sensitivity metric based on gradients to fine-tune the high-sensitivity parameters. Prompt-based methods incorporate visual prompt tokens (Jia et al., 2022) into the downstream inputs, which then undergo parameter updates. Various prompt structures (Das et al., 2023; Zhou et al., 2024) and parameter-efficient strategies (Han et al., 2023) are explored to further enhance performance. Adapter-based approaches introduce additional lightweight adaptation networks rather than modifying inputs. AdaptFormer (Chen et al., 2022) and LoRand (Yin et al., 2023) incorporate adaptation modules with bottleneck structures and residual connections, while LoRA (Hu et al., 2022) and RLRR (Dong et al., 2024a) employ learnable low-rank parameter bypasses to facilitate adaptation.

More recently, some efforts apply PETL methods to more complex tasks like segmentation (Yin et al., 2023) or to leverage stronger pre-trained models such as SAM (Kirillov et al., 2023). However, despite achieving promising results in fields like medical image segmentation (Wu et al., 2023) and camouflaged object detection (Chen et al., 2023), a notable performance gap persists between PETL and full fine-tuning in semantic segmentation tasks, particularly when compared to the successes realized in image classification.

### 2.2 PROGRESSIVE DESIGN IN TRANSFER LEARNING

Progressive paradigm is a prevalent concept in transfer learning (Rusu et al., 2016; Weinshall et al., 2018), aimed at enhancing model performance by gradually adapting to the target domain. This approach is particularly advantageous when there is a substantial gap between the source and target domains (Hsu et al., 2020), facilitating smoother transitions and more effective knowledge transfer. For instance, Progressive Neural Networks (Rusu et al., 2016) introduce new columns of neurons for each new reinforcement learning task while keeping the parameters of the previous columns fixed. In the field of domain adaptation, Hsu et al. (2020) generate intermediate domain data situated between the source and target domains using CycleGAN (Zhu et al., 2017), progressively addressing domain adaptation challenges. Similarly, PPEA (Dong et al., 2024b) designs a progressive training framework tailored for depth estimation that utilizes external datasets to construct an easy-to-hard pipeline, iteratively updating the adapter at various training stages.

Despite the success of these approaches in mitigating data-domain shifts, extending the progressive paradigm from adaptation across similar tasks in different data domains to addressing the fine-grained perceptual requirements of downstream segmentation remains underexplored.

## 3 METHOD

### 3.1 VANILLA PETL

Given a pre-trained model $f_{\boldsymbol{\theta}}(\cdot)$ parameterized by $\boldsymbol{\theta}$, the goal of PETL is to efficiently adapt this pre-trained model to a downstream task, yielding a model $f_{\boldsymbol{\theta},\phi}(\cdot)$, where $\phi$ denotes the weights of the adaptation module. During the adaptation process, with the training set $\boldsymbol{D} = \{(\boldsymbol{x}_i, \boldsymbol{y}_i)\}_{i=1}^{N}$ from the downstream task, PETL freezes $\boldsymbol{\theta}$ and only updates $\phi$ by optimizing the following objective:

$$\phi^* = \arg\min_{\phi} \mathcal{L}\left(f_{\boldsymbol{\theta},\phi}(\cdot);\ \boldsymbol{D}, \phi^r\right),  \tag{1}$$

where $\mathcal{L}(\cdot)$ denotes the loss function for the downstream task and $\phi^r$ indicates random initialization of the parameters in the adaptation module.

By fully leveraging the pre-training weights of the pre-trained model, PETL achieves the same or even better performance than full fine-tuning by updating only $\phi$. Since the number of parameters in

Figure 2: Illustration of two progressive adaptation strategies. Left: Generalized Parametric Adaptation. Right: Decoupled Structured Adaptation. "FFN" indicates the Feed-Forward Network.

$\phi$ is only a small fraction of the total model parameters $\boldsymbol{\theta}$, PETL can significantly enhance computational efficiency and reduce the cost of the fine-tuning process while improving accuracy compared to full fine-tuning. However, PETL encounters bottlenecks when applied to downstream semantic segmentation tasks due to changes in perceptual granularity.

## 3.2 MIDSTREAM PROGRESSIVE ADAPTATION

To mitigate semantic granularity shifts, we divide the adaptation into two phases: midstream adaptation and downstream fine-tuning. By introducing these additional training phases, the adaptation can be enhanced with supervision from intermediate tasks, thereby facilitating a better transition of the pre-trained model into downstream tasks. For this purpose, we investigate two candidate progressive adaptation strategies: Generalized Parametric Adaptation (GPA) and Decoupled Structured Adaptation (DSA), which are graphically illustrated in Fig. 2.

**Generalized Parametric Adaptation** is a straightforward progressive adaptation strategy that utilizes enhanced supervision during the midstream phase to bridge perception gaps between the pre-trained model and downstream tasks. Formally, during the midstream adaptation phase, the pre-trained weights $\boldsymbol{\theta}$ are frozen and only $\phi$ is updated by optimizing the following objective:

$$\phi^m = \arg\min_{\phi} \mathcal{L}^m \left( f_{\boldsymbol{\theta}, \phi}(\cdot); \ \boldsymbol{D}^m, \phi^{init} \right), \tag{2}$$

where $\mathcal{L}^m$ and $\boldsymbol{D}^m$ denote the loss function and training data of the intermediate task, $\phi^{init}$ denotes the initial weights and $\phi^m$ indicates the optimized weights of $\phi$. In the second training phase, GPA begins to fit $\phi$ starting from $\phi^m$ using the downstream data and objective, formalized as:

$$\phi^d = \arg\min_{\phi} \mathcal{L} \left( f_{\boldsymbol{\theta}, \phi}(\cdot); \boldsymbol{D}, \phi^m \right). \tag{3}$$

With the improved initialization from the midstream adaptation, $\phi$ tends to generalize better in the downstream fine-tuning phase compared to vanilla PETL methods.

For the adaptation module, GPA introduces a bypass branch to the Feed-Forward Network (FFN) layer. This bypass branch uses a bottleneck structure with a down-projection layer and an up-projection layer, parameterized by $\boldsymbol{W}_{\text{down}} \in \mathbb{R}^{D \times \frac{D}{r}}$ and $\boldsymbol{W}_{\text{up}} \in \mathbb{R}^{\frac{D}{r} \times D}$, respectively. $D$ denotes the channel dimension and $r$ is the reduction factor for parameter efficiency. For the input feature $\boldsymbol{z}$ of the FFN layer, the adapted feature $\tilde{\boldsymbol{z}}$ is obtained by adding the output of the bypass branch to the output of the FFN layer, formalized as:

$$\tilde{\boldsymbol{z}} = \text{FFN}(\boldsymbol{z}) + \text{ReLU}(\boldsymbol{z} \cdot \boldsymbol{W}_{\text{down}}) \cdot \boldsymbol{W}_{\text{up}}. \tag{4}$$

During midstream adaptation, GPA optimizes the parameters within the bypass branch according to Eq. 2 and keeps the optimized parameters $\{\boldsymbol{W}_{\text{down}}^m, \boldsymbol{W}_{\text{up}}^m\}$. In downstream fine-tuning, GPA uses the same structure as in the midstream adaptation phase (refer to Eq. 4) and initializes $\{\boldsymbol{W}_{\text{down}}, \boldsymbol{W}_{\text{up}}\}$ with $\{\boldsymbol{W}_{\text{down}}^m, \boldsymbol{W}_{\text{up}}^m\}$ and further optimizes them based on the downstream task loss.

**Decoupled Structured Adaptation** is another progressive adaptation strategy for PETL. Unlike GPA, which updates the same adapter for both intermediate and downstream tasks, DSA splits the adaptation parameters into two parts $\{\phi, \psi\}$. During the midstream adaptation phase, only $\phi$ will be optimized. In the downstream fine-tuning phase, $\phi$ is frozen, and optimization is focused solely on $\psi$. By successively optimizing the following objective:

$$\phi^m = \arg\min_{\phi} \mathcal{L}^m \left( f_{\boldsymbol{\theta}, \phi}(\cdot); \boldsymbol{D}^m, \phi^{init} \right), \tag{5}$$

$$\psi^d = \arg\min_{\psi} \mathcal{L} \left( f_{\boldsymbol{\theta}, \phi, \psi}(\cdot); \boldsymbol{D}, \phi^m, \psi^{init} \right), \tag{6}$$

DSA thus obtains the optimized parameters $\phi^m$ and $\psi^d$ during the two-phase training process. The key difference between DSA and GPA lies in the decoupled weights. By freezing part of the adaptation parameters $\phi$ after midstream adaptation, DSA is able to avoid the forgetting problem that can occur during downstream fine-tuning in GPA. Refer to Section 4.3 for further analysis.

Specifically, during the midstream adaptation, DSA introduces a bypass branch to the FFN layer, which shares the same structure as in Eq. 4, and obtains the optimized parameters $\{\boldsymbol{W}_{\text{down}}^m, \boldsymbol{W}_{\text{up}}^m\}$ according to Eq 5. In the downstream fine-tuning stage, DSA first *recovers* the feature $\dot{\boldsymbol{z}}$ after midstream adaptation by applying:

$$\dot{\boldsymbol{z}} = \text{FFN}(\boldsymbol{z}) + \text{ReLU}(\boldsymbol{z} \cdot \boldsymbol{W}_{\text{down}}) \cdot \boldsymbol{W}_{\text{up}}, \tag{7}$$

where the projection layer parameters $\{\boldsymbol{W}_{\text{down}}, \boldsymbol{W}_{\text{up}}\}$ are initialized with $\{\boldsymbol{W}_{\text{down}}^m, \boldsymbol{W}_{\text{up}}^m\}$ and kept frozen. To address potential task bias introduced during midstream adaptation, DSA concatenates $\dot{\boldsymbol{z}}$ with the original FFN output along the channel dimension and feeds them into a randomly initialized bypass branch parameterized by $\{\hat{\boldsymbol{W}}_{\text{down}} \in \mathbb{R}^{2D \times \frac{2D}{r}}, \hat{\boldsymbol{W}}_{\text{up}} \in \mathbb{R}^{\frac{2D}{r} \times D}\}$. This allows $\{\hat{\boldsymbol{W}}_{\text{down}}, \hat{\boldsymbol{W}}_{\text{up}}\}$ to selectively integrate midstream information and progressively adapt intermediate features for downstream tasks. DSA obtains the final adapted feature by adding the bypass branch output to the FFN output and optimizes $\{\hat{\boldsymbol{W}}_{\text{down}}, \hat{\boldsymbol{W}}_{\text{up}}\}$ according to Eq. 6. The above process is formalized as:

$$\tilde{\boldsymbol{z}} = \text{FFN}(\boldsymbol{z}) + \text{ReLU}\left(\left[\dot{\boldsymbol{z}}, \text{FFN}(\boldsymbol{z})\right] \cdot \hat{\boldsymbol{W}}_{\text{down}}\right) \cdot \hat{\boldsymbol{W}}_{\text{up}}, \tag{8}$$

where $[\cdot, \cdot]$ indicates the channel concatenation operator.

### 3.3 INTERMEDIATE TASK DESIGNATION

In progressive transfer learning for semantic segmentation, the intermediate task is expected to enhance the model's fine-grained perception to produce dense predictions for diverse classes, as required by downstream applications. Two key factors should be considered in this context, *i.e.,* perception granularity and supervision diversity.

**Perception Granularity.** Unlike pre-trained tasks focused on global perception for object-level classification, segmentation tasks require much finer perception granularity to capture pixel-level semantics. The intermediate task should help the model bridge this shift in granularity. To achieve this, we design a transfer strategy that converts the downstream training data with dense annotations to support training at varying levels of perception granularity.

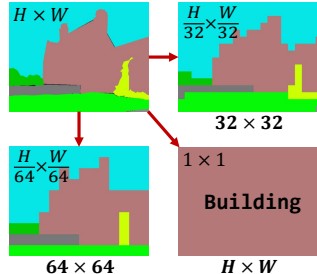

Specifically, for the ground truth dense annotation map $\boldsymbol{y} \in \mathbb{R}^{H \times W}$ from the downstream dataset $\boldsymbol{D}$, where $H$ and $W$ indicate the image height and width, we obtain its one-hot label vector $\hat{\boldsymbol{y}} \in \mathbb{R}^{H \times W \times C}$, where $C$ is the total number of categories in $\boldsymbol{D}$. The perception granularity transformation $F_{\text{PG}}(\cdot)$ is performed by down-

Figure 3: Illustration of $F_{\text{PG}}$.

sampling $\hat{\boldsymbol{y}}$ using pooling windows of size $s \times s$ with a stride of $s$, thereby generating annotations at different levels of granularity by controlling $s$.

For generating image-level labels, $F_{\text{PG}}(\cdot)$ applies an $H \times W$ pooling window, producing a single label vector $F_{\text{PG}}(\hat{\boldsymbol{y}}) \in \mathbb{R}^C$ for the entire input image. For patch-level labels, $F_{\text{PG}}(\cdot)$ sets $s$ to match the patch size of the pre-trained transformer, resulting in $F_{\text{PG}}(\hat{\boldsymbol{y}}) \in \mathbb{R}^{\frac{H}{s} \times \frac{W}{s} \times C}$, which yields $\frac{H}{s} \times \frac{W}{s}$ label vectors, each corresponding to a distinct patch region. Fig. 3 visualizes the output of $F_{\text{PG}}(\hat{\boldsymbol{y}})$ with varying pooling window size. Intuitively, after applying $F_{\text{PG}}(\cdot)$, the intermediate task requires the model to perform coarse-grained region classification rather than classifying each input pixel individually.

**Supervision Diversity.** In addition to shifts in perception granularity, downstream segmentation tasks require classification pre-trained models to develop a broader semantic awareness beyond recognizing a single dominant class. To address this gap, we incorporate semantic diversity modeling into the intermediate task design by introducing supervision diversity transformation $F_{\text{SD}}(\cdot)$.

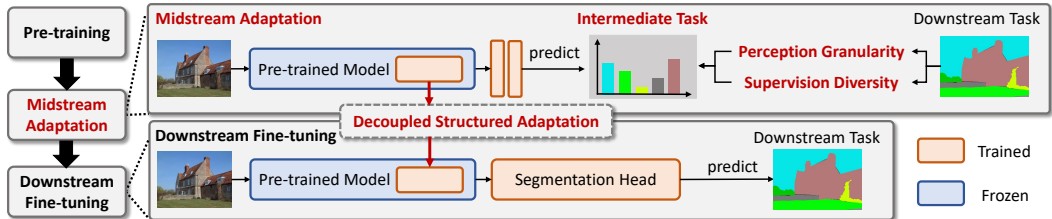

Figure 5: Illustration of the ProPETL framework. In the midstream phase, we generate the midstream dataset via perception granularity and supervision diversity transformations and update the adaptation module. In the downstream fine-tuning phase, we apply the decoupled structured adaptation strategy and train both the adaptation module and segmentation head on the downstream dataset.

To be specific, $F_{\mathrm{SD}}(\cdot)$ refines the pooling method utilized in $F_{\mathrm{PG}}(\cdot)$ by employing max-pooling to generate one-hot labels $\hat{\boldsymbol{y}}_{\mathrm{o}}$ and mean-pooling to create many-hot labels $\hat{\boldsymbol{y}}_{\mathrm{m}}$, which reflect the semantic density within a given region. To finely control supervision diversity and inspired by label smoothing (Szegedy et al., 2016), $F_{\mathrm{SD}}(\cdot)$ interpolates between $\hat{\boldsymbol{y}}_{\mathrm{o}}$ and $\hat{\boldsymbol{y}}_{\mathrm{m}}$ using the equation $(1-\alpha)*\hat{\boldsymbol{y}}_{\mathrm{o}}+\alpha*\hat{\boldsymbol{y}}_{\mathrm{m}}$, where $\alpha \in [0,1]$ serves as a smoothing factor. Fig. 4 illustrates the output of $F_{\mathrm{SD}}(\hat{\boldsymbol{y}})$ with varying $\alpha$ when $s = H \times W$. By varying $\alpha$, $F_{\mathrm{SD}}(\cdot)$ generates labels with different numbers of categories, thus modulating the supervision diversity of the intermediate tasks.

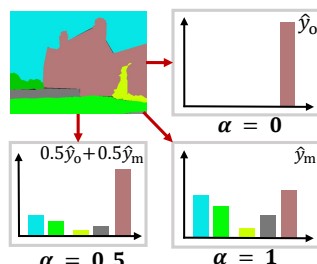

Figure 4: Illustration of $F_{\mathrm{SD}}$.

By combining these two factors, we formulate intermediate tasks that require predicting the generated labels $F_{\mathrm{SD}} \circ F_{\mathrm{PG}}(\hat{\boldsymbol{y}}_i)$ from input $\boldsymbol{x}_i$. This approach enhances the pre-trained model's fine-grained perception ability, helping it better adapt to downstream segmentation applications.

## 3.4 FRAMEWORK AND COMPLEXITY

**Framework.** The entire framework is illustrated in Fig. 5. The training process of ProPETL consists of two phases: midstream adaptation and downstream fine-tuning.

In the midstream adaptation stage, we transform the downstream dataset $\boldsymbol{D}$ with respect to perception granularity and supervision diversity to obtain an intermediate dataset $\boldsymbol{D}^m$. To prevent overfitting on the intermediate task, we use a two-layer MLP as the intermediate task head, following standard practices (Chen et al., 2020a;b). We evaluate different combinations of perception granularity and supervision diversity, and based on these empirical studies, we employ image-level granularity ($s = H \times W$) and multi-label diversity ($\alpha = 1$) for midstream adaptation in our framework. The standard cross-entropy loss, denoted as $\mathcal{L}^m(\cdot)$, is used to optimize both the bypass branch and the intermediate task head. In the downstream fine-tuning stage, ProPETL first processes adaptation module parameters according to the decoupled structured adaptation strategy. Subsequently, the downstream segmentation head is randomly initialized, and the learnable parameters are fine-tuned with the downstream dataset. For inference, the trained adaptation module, combined with the pre-trained backbone, extracts feature maps from input images, which are then fed to the segmentation head to generate the final segmentation results.

**Parametric Complexity.** When employing GPA as the progressive adaptation strategy, the number of learnable parameters in the backbone network is $O(\frac{2LD^2}{r})$, where $L$ denotes the number of backbone layers. In contrast, DSA increases the number of learnable parameters to $O(\frac{8LD^2}{r})$ due to the incorporation of additional structures. To maintain parameter consistency, we empirically set the reduction factor $r$ in DSA to be four times that used in GPA.

Table 1: Comparative results on four semantic segmentation datasets. The experiments utilize an ImageNet-21k supervised pre-trained Swin-L as the backbone and UperNet as the segmentation head. The best and second-best results of the PETL method are highlighted in **bold** and underline, respectively. The symbol "*" denotes the learnable parameters in the backbone.

| Method | VOC2012 | | ADE20k | | COCO-Stuff10k | | CityScapes | | Mean | | Learnable |
| | mIoU | mAcc | mIoU | mAcc | mIoU | mAcc | mIoU | mAcc | mIoU | mAcc | Param.*(M) |
|---|---|---|---|---|---|---|---|---|---|---|---|
| Full fine-tuning | 84.38 | 89.82 | 51.71 | 63.13 | 46.30 | 58.98 | 82.36 | 88.53 | 66.19 | 75.12 | 195.00 |
| Freeze | 83.32 | 89.16 | 47.51 | 59.74 | 42.36 | 54.39 | 75.54 | 82.91 | 62.18 | 71.55 | **0** |
| BitFit | 84.25 | 90.25 | 48.00 | 60.81 | 43.75 | 56.37 | 77.73 | 84.89 | 63.43 | 73.08 | 0.30 |
| LoRand | 84.09 | 90.32 | 48.08 | 59.31 | 43.96 | 56.41 | 76.88 | 83.89 | 63.25 | 72.48 | 3.59 |
| RLRR | 84.77 | 90.72 | 48.72 | 60.53 | 44.51 | 57.03 | 78.43 | 85.35 | 64.11 | 73.41 | 0.46 |
| VPT | 85.69 | 91.12 | 50.35 | 61.37 | 45.04 | 57.58 | 78.95 | 86.56 | 65.01 | 74.16 | 3.61 |
| AdaptFormer | 85.41 | 90.93 | 50.38 | 62.55 | 45.45 | 57.83 | 79.41 | 86.03 | 65.16 | 74.34 | 2.64 |
| LoRA | 85.36 | 91.34 | 50.45 | 62.91 | 45.03 | 57.81 | 80.54 | 87.23 | 65.35 | 74.82 | 4.55 |
| ProPETL | **86.11** | **92.08** | **51.05** | **63.61** | **46.48** | **58.69** | 81.67 | 88.23 | **66.33** | **75.65** | 3.30 |

# 4 EXPERIMENTS

## 4.1 EXPERIMENTAL SETUP

**Datasets.** We conduct a comprehensive evaluation of the proposed ProPETL for segmentation adaptation on benchmarks, including PASCAL VOC2012 (Everingham et al., 2015), ADE20k (Zhou et al., 2019), COCO-Stuff10k (Caesar et al., 2018), and CityScapes (Cordts et al., 2016). **1) PASCAL VOC2012** consists of 1,464 training images and 1,449 validation images spanning 21 categories. It is widely adopted in object detection and semantic segmentation, making it a key benchmark for tasks involving limited-scale data. **2) ADE20k** includes 20,210 training images and 2,000 validation images, making it one of the most comprehensive datasets for semantic segmentation with a wide range of data variances. **3) COCO-Stuff10k** extends the COCO dataset (Lin et al., 2014) with dense annotations for semantic segmentation, including 9,000 training images and 1,000 validation images across 172 categories. **4) CityScapes** includes finely annotated images across 19 categories, with 2,975 for training and 500 for validation. It is widely used for evaluating urban scene understanding.

**Implementation details.** We employ AdamW (Loshchilov & Hutter, 2018) for optimization, with $\beta_1$ and $\beta_2$ set to 0.9 and 0.999, respectively. A PolyLR scheduler is used to dynamically adjust the learning rate during the training process. The batch size is set to 16 across all benchmarks, and the iterations for downstream fine-tuning range from 40k to 160k, consistent with those in previous works (Xiao et al., 2018; Liu et al., 2021) for a fair comparison. For midstream adaptation, the iterations are set to half of the corresponding downstream settings.

Following the counterpart (Yin et al., 2023), we utilize Swin-L (Liu et al., 2021), pre-trained on ImageNet-21K (Deng et al., 2009), as the backbone, and UperNet (Xiao et al., 2018) as the segmentation framework. The mean intersection over union (mIoU) and mean accuracy (mAcc) on the validation set across all benchmarks are used as metrics to provide a comprehensive comparison with state-of-the-art methods. All experiments are implemented using MMSegmentation (Contributors, 2020) on NVIDIA A800 GPU.

## 4.2 COMPARISON WITH THE STATE-OF-THE-ARTS

We compare the proposed ProPETL with its counterparts on segmentation benchmarks, including full fine-tuning, freeze (*a.k.a.* linear probing), and several typical PETL methods. The PETL counterparts include partial tuning methods like Bias (Zaken et al., 2022), prompt-based methods like VPT (Jia et al., 2022), and adapter-based methods such as AdaptFormer (Chen et al., 2022), LoRand (Yin et al., 2023), LoRA (Hu et al., 2022), and RLRR (Dong et al., 2024a).

As illustrated in Table 1, ProPETL delivers highly competitive performance across all settings, significantly surpassing PETL counterparts by a large margin in both metrics. Notably, on the challenging COCO-Stuff10k benchmark, it outperforms the previous state-of-the-art, AdaptFormer, by

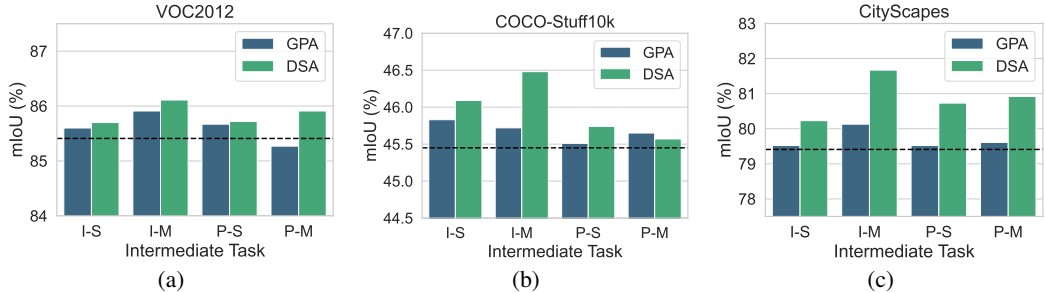

Figure 6: Ablation results of two progressive adaptation strategies. The dashed line represents the result of the baseline approach, *i.e.* AdaptFormer. "I" and "P" denote that the perception granularity of the intermediate task is image-level and patch-level, respectively. "S" and "M" indicate that the supervision diversity of intermediate task is single-label and multi-label, respectively.

1.03% mIoU. Additionally, ProPETL exceeds full fine-tuning while using only 1.7% of the parameters. To the best of our knowledge, this is the first time a PETL method has outperformed full fine-tuning on this benchmark, demonstrating its superiority. On the most challenging benchmark, ADE20K, the proposed method also delivers performance comparable to full fine-tuning and even surpasses it in the mAcc metric, indicating its generalizability across various segmentation scenarios.

When considering parameter efficiency, it is worth noting that although LoRand, VPT, and LoRA utilize a comparable amount of learnable parameters as ProPETL, their performances are significantly inferior. This phenomenon demonstrates that the introduction of the progressive paradigm effectively enhances the adaptation ability to manage the semantic perception granularity shift in downstream segmentation tasks within PETL, while also maintaining computational efficiency.

## 4.3 ABLATION STUDY

**Progressive Adaption Strategy.** We conduct detailed ablation studies on two typical progressive adaptation strategies with representative intermediate tasks on the VOC2012, COCO-Stuff10k and CityScapes benchmarks. The results are illustrated in Fig. 6. From Fig. 6, we observe that, compared to the AdaptFormer baseline, the introduction of the progressive paradigm significantly boosts performance in most settings, demonstrating its effectiveness for semantic segmentation adaptation. Moreover, DSA consistently outperforms GPA in most settings, highlighting its superiority.

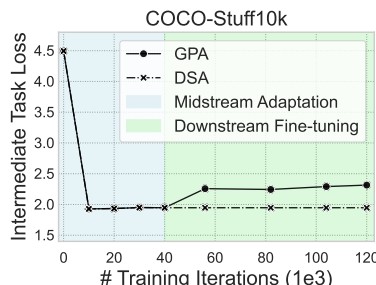

Figure 7: Visualization of the loss curves for the intermediate task.

We further investigate this phenomenon and visualize the loss function curve during training in Fig. 7. As shown in the figure, when training with downstream tasks, GPA tends to forget a portion of the knowledge learned during the intermediate tasks, as indicated by the increasing loss value of the intermediate task. In contrast, since DSA freezes parts of the adapter after midstream adaptation, it continues to transfer effectively to the downstream task without degradation, thereby delivering better performance. Based on these analysis, we employ DSA in our framework.

**Intermediate Task Design.** We conduct ablation studies to investigate the impact of perception granularity and supervision diversity. To mitigate perturbations, we average results from experiments with varying supervision diversities at each perception granularity level, and vice versa [1]. As shown in Table 2, the VOC2012 and COCO-Stuff10k datasets exhibit a preference for intermediate task with image-level perception, while the CityScapes dataset favors more fine-grained intermediate tasks. This divergence can be attributed to the fact that VOC2012 and COCO-Stuff10k contain a greater number of object-centric images; the image-level intermediate task necessitates perceiving the objects existing in the image, thereby providing a basis for the downstream segmentation tasks.

---

[1] Please refer to Table 13 in the Appendix for full results.

Table 2: Ablation study on the perception granularity of intermediate tasks using DSA during downstream fine-tuning. "$s$" denotes the pooling window size. The first line is the result of AdaptFormer.

| Patch-level | $\longleftrightarrow$ | Image-level | VOC2012 | | COCO-Stuff10k | | CityScapes | |
|---|---|---|---|---|---|---|---|---|
| $s = 32$ | $s = 128$ | $s = 640$ | mIoU | mAcc | mIoU | mAcc | mIoU | mAcc |
| | | | 85.41 | 90.93 | 45.45 | 57.83 | 79.41 | 86.03 |
| ✓ | | | 85.75 | 91.30 | 45.60 | 57.80 | 80.84 | 87.26 |
| | ✓ | | 85.62 | **92.04** | 45.99 | 58.04 | **81.03** | **87.58** |
| | | ✓ | **85.89** | 91.68 | **46.32** | **58.33** | 80.89 | 87.54 |

Table 3: Ablation study on the supervision diversity of intermediate tasks using DSA during downstream fine-tuning. "$\alpha$" represents the smoothing factor. The first line is the result of AdaptFormer.

| Single label | $\longleftrightarrow$ | Multi label | VOC2012 | | COCO-Stuff10k | | CityScapes | |
|---|---|---|---|---|---|---|---|---|
| $\alpha = 0$ | $\alpha = 0.5$ | $\alpha = 1$ | mIoU | mAcc | mIoU | mAcc | mIoU | mAcc |
| | | | 85.41 | 90.93 | 45.45 | 57.83 | 79.41 | 86.03 |
| ✓ | | | 85.65 | 91.59 | 45.86 | 58.08 | 80.56 | 87.06 |
| | ✓ | | 85.68 | 91.39 | 46.00 | 57.75 | 80.89 | 87.40 |
| | | ✓ | **85.94** | **92.05** | **46.05** | **58.33** | **81.31** | **87.92** |

Conversely, the images in CityScapes typically feature complete traffic scenes with relatively fixed semantic categories, limiting the diversity of image-level labels. Hence, fine-grained intermediate tasks are preferred. When examining supervision diversity in Table 3, we observe that the multi-label intermediate task consistently outperforms the single-label task. This advantage may stem from the single-label task's focus on perceiving only the most frequent semantic categories within a region, thereby neglecting other important semantic information and complicating adaptation to downstream pixel-level segmentation tasks.

Overall, the optimal performance is achieved when using a combination of image-level perception, where the pooling window size is set equal to the input image size ($640 \times 640$) and multi-label supervision, with the smoothing factor $\alpha$ set to 1. Based on this empirical observation, we adopt this combination in our framework.

## 4.4 GENERALIZATION EVALUATION.

To evaluate the generalization ability of the proposed ProPETL, we evaluate it with different pre-trained models and downstream tasks. In these comparisons, we consider AdaptFormer, one of the state-of-the-art methods, as a counterpart since it shares a similar framework with ProPETL, excluding the progressive paradigm.

**Pre-trained models.** To assess ProPETL's versatility across diverse pre-training paradigms, we extend the evaluation to models with self-supervised pre-training. Specifically, Table 4 presents results for MAE, a self-supervised model based on Masked Image Modeling (MIM) pre-training, while Table 5 shows results for MoCo-v3, a self-supervised model that utilizes contrastive learning. As shown in Tables 4 and 5, even when using a less powerful backbone such as ViT-B/16 (Dosovitskiy et al., 2020), ProPETL still significantly outperforms its counterparts. Notably, on the challenging CityScapes dataset, ProPETL surpasses AdaptFormer by a large margin of 7% mIoU based on MAE, demonstrating its robust generalization ability across different pre-trained models. Additionally, we evaluate SAM (Kirillov et al., 2023), a model specifically pre-trained for mask segmentation. As shown in Table 6, ProPETL consistently outperforms AdaptFormer and the Freeze baseline when using the SAM backbone, although it slightly lags behind full fine-tuning in certain cases.

**Downstream tasks.** When addressing a more complex segmentation task, *i.e.,* instance segmentation, we further calculate the average precision on CityScapes in Table 7. The improvements are even more pronounced in this context, with ProPETL substantially outperforming the counterparts, highlighting its potential for broader application in other challenging computer vision tasks.

Table 4: Comparison of results using ViT-B/16 pre-trained by MAE in a self-supervised manner as the backbone and UperNet as the segmentation head.

| | COCO-Stuff10k | | CityScapes | | Param. |
|---|---|---|---|---|---|
| Method | mIoU | mAcc | mIoU | mAcc | * (M) |
| Full ft. | 39.64 | 51.64 | 80.87 | 87.79 | 87.02 |
| Freeze | 28.94 | 39.58 | 62.22 | 69.96 | **0** |
| AdaptFormer | 34.42 | 45.60 | 70.99 | 78.75 | 1.19 |
| ProPETL | **37.03** | **49.19** | **77.75** | **85.25** | 1.19 |

Table 5: Comparison of results using ViT-B/16 pre-trained by MoCo-v3 in a self-supervised manner as the backbone and UperNet as the segmentation head.

| | COCO-Stuff10k | | CityScapes | | Param. |
|---|---|---|---|---|---|
| Method | mIoU | mAcc | mIoU | mAcc | * (M) |
| Full ft. | 18.00 | 26.05 | 64.58 | 73.26 | 85.84 |
| Freeze | 10.10 | 15.60 | 42.02 | 48.47 | **0** |
| AdaptFormer | 14.69 | 22.33 | 54.93 | 62.86 | 1.19 |
| ProPETL | **15.48** | **23.04** | **58.83** | **67.00** | 1.19 |

Table 6: Comparison of results using SAM-Large as the backbone and Mask2Former as the segmentation head.

| | CityScapes | | Param. |
|---|---|---|---|
| Method | mIoU | mAcc | * (M) |
| Full ft. | 82.38 | 89.96 | 305.58 |
| Freeze | 67.04 | 79.31 | **0** |
| AdaptFormer | 79.02 | 88.05 | 6.32 |
| ProPETL | **80.24** | **89.10** | 6.32 |

Table 7: Instance segmentation results using supervised pre-trained Swin-L as the backbone and Mask R-CNN as the segmentation head.

| | CityScapes | | | | Param. |
|---|---|---|---|---|---|
| Method | $AP^{box}$ | $AP_{50}^{box}$ | $AP^{mask}$ | $AP_{50}^{mask}$ | * (M) |
| Full ft. | 41.60 | 69.40 | 38.20 | 64.50 | 195.00 |
| Freeze | 9.60 | 21.70 | 8.80 | 18.80 | **0** |
| AdaptFormer | 23.10 | 45.80 | 21.50 | 42.00 | 2.65 |
| ProPETL | **30.00** | **58.50** | **29.30** | **54.00** | 2.65 |

## 5    CONCLUSION

In this paper, we propose ProPETL, a progressive PETL paradigm designed to adapt pre-trained models to downstream segmentation tasks. By introducing an additional training phase in the fine-tuning process, ProPETL successfully enhances fine-grained perception ability and improves performance on downstream segmentation tasks. Various progressive adaptation strategies and intermediate task designs are comprehensively explored to achieve optimal transfer effectiveness. Furthermore, extensive comparisons demonstrate that ProPETL delivers superior performance across benchmarks, underscoring its effectiveness and superiority.

**Limitations and Future Works.** While ProPETL demonstrates significant improvements in performance, it has several limitations. On the one hand, while the midstream adaptation phase enhances fine-tuning performance, it does increase the overall computational cost of training[2]. Exploring methods to shorten the global fine-tuning process and further improve fine-tuning efficiency remains an open challenge. On the other hand, the current design of intermediate tasks primarily relies on empirical studies. Developing dynamic approaches for designing intermediate tasks and generalizing ProPETL to accommodate a broader range of pre-trained models and downstream scenarios are promising directions for future research.

**Acknowledgments.** The authors sincerely thank the anonymous reviewers for their insightful comments and constructive suggestions. This work is partly supported by the National Natural Science Foundation of China (82441024), the Research Program of State Key Laboratory of Critical Software Environment, and the Fundamental Research Funds for the Central Universities.

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

# A    APPENDIX

In this section, we present an analysis of computational overhead, visualization results, and complete ablation study results on intermediate task design. Additionally, we provide extended evaluations using a broader range of pre-trained models and segmentation heads.

**Computational Overhead.** We compare the computational overhead— including training time, inference time, inference workload, and GPU memory footprint— of the proposed ProPETL against its counterparts in Table 8. To ensure a fair comparison, we use Swin-L as the backbone and train all methods for 80,000 iterations with a batch size of 16 on two NVIDIA A800 GPU. ProPETL consumes only 55GB of GPU memory during training and 627 GFLOPs during inference, significantly less than both VPT and LoRA. While its GPU memory usage and parameter count are slightly higher than AdaptFormer's, ProPETL outperforms it substantially, as shown in Table 1. To further reduce computational costs, we introduce a lighter version of ProPETL that minimizes the fine-tuning process to achieve the shortest training time. The comparisons are detailed in Table 9.

Table 8: Comparison of the computational overhead. "$\dagger$" indicates the short training version. Note: reparameterization for LoRA was not implemented.

| Method | Training time↓ (Hour) | Inference time↓ (Millisecond) | Workload ↓ (GFLOPs) | GPU Memory↓ (MB) | Param.↓ (M) |
|---|---|---|---|---|---|
| Full fine-tuning | 41 | **182** | **623.310** | 94,198 | 195.00 |
| VPT | 96 | 210 | 872.222 | 135,498 | 3.61 |
| LoRA | 30 | 208 | 718.097 | 94,564 | 4.55 |
| AdaptFormer | 34 | 185 | 627.085 | **51,433** | **2.64** |
| ProPETL | 47 | 191 | 627.111 | 55,613 | 3.30 |
| ProPETL$^{\dagger}$ | **29** | 191 | 627.111 | 55,613 | 3.30 |

As shown in Table 9, although AdaptFormer uses the same total number of iterations as ProPETL$^{\dagger}$, the latter significantly outperforms AdaptFormer on both the COCO-Stuff10k and CityScapes benchmarks, thanks to its progressive adaptation strategy. Additionally, we extend AdaptFormer's iterations to 120k for a more comprehensive comparison with ProPETL. However, this extension only slightly improves AdaptFormer's performance, as it has already saturated in mitigating the perceptual granularity shifts toward downstream tasks. In contrast, ProPETL shows substantial performance improvements, as its midstream task convergence enables pre-trained models to better adapt to downstream segmentation scenarios, further demonstrating the effectiveness of the proposed strategy.

Table 9: Ablation study on the training iterations. "$\dagger$" indicates the short training version.

| Method | COCO-Stuff10k | | CityScapes | | Midstream Adaptation | Downstream Fine-tuning | Total Iterations |
|---|---|---|---|---|---|---|---|
| | mIoU | mAcc | mIoU | mAcc | | | |
| AdaptFormer | 45.45 | 57.83 | 79.41 | 86.03 | N/A | 80,000 | 80,000 |
| ProPETL$^{\dagger}$ | **45.98** | **58.20** | **80.93** | **87.51** | 40,000 | 40,000 | 80,000 |
| AdaptFormer | 45.81 | 58.13 | 79.78 | 86.42 | N/A | 120,000 | 120,000 |
| ProPETL | **46.48** | **58.69** | **81.67** | **88.23** | 40,000 | 80,000 | 120,000 |

**More Generalization Evaluation.** We evaluate the effectiveness of ProPETL across more pre-trained models and segmentation heads. In the main body, we report results using pre-trained Swin-transformer-large as the backbone. Additionally, Table 10 presents an analysis of ProPETL on the VOC2012 dataset using different sizes of Swin Transformers. These results consistently show that ProPETL outperforms both AdaptFormer and full fine-tuning, underscoring its effectiveness even with smaller pre-trained models. To further assess ProPETL's versatility across diverse pre-training paradigms, we extend the evaluation to models with self-supervised pre-training. Table 11 reports results using DINOv2, a self-supervised model built upon an extended contrastive learning framework. For segmentation heads, we evaluate the performance of Mask2Former (Cheng et al., 2022), which employs mask-based prediction strategies. Notably, Table 11 shows that when paired

with DINOv2 as the backbone and Mask2Former as the segmentation haed, ProPETL surpasses both full fine-tuning and AdaptFormer, further highlighting its potential as a flexible and high-performing PETL framework.

Table 10: Comparative results on VOC2012 using different size of Swin-transformers with UperNet.

| Method | Swin-small | | Swin-base | | Swin-large | |
|---|---|---|---|---|---|---|
| | mIoU | mAcc | mIoU | mAcc | mIoU | mAcc |
| Full fine-tuning | 80.83 | 86.90 | 82.07 | 87.63 | 84.38 | 89.82 |
| AdaptFormer | 80.94 | 87.26 | 84.29 | 90.31 | 85.41 | 90.83 |
| ProPETL | **81.26** | **87.82** | **85.03** | **90.96** | **86.11** | **92.08** |

Table 11: Comparative results of using DINOv2 with Mask2Former.

| Method | VOC2012 | | CityScapes | | Param. $^*$ (M) |
|---|---|---|---|---|---|
| | mIoU | mAcc | mIoU | mAcc | |
| Full fine-tuning | 86.81 | 93.35 | 84.10 | 90.57 | 304.19 |
| AdaptFormer | 88.93 | 94.34 | 83.83 | 90.87 | 4.74 |
| ProPETL | **89.53** | **94.79** | **84.29** | **91.37** | 4.74 |

**Cross-scenario Evaluation**. We conducted cross-scenario evaluation of ProPETL under the domain generalization setting and compared it against Full fine-tuning and AdaptFormer. We additionally compare to Rein(Wei et al., 2024), which is a recent parameter efficient method for domain generalization semantic segmentation. For a fair comparation, we utilized the DINOv2(Oquab et al., 2023) + Mask2Former(Cheng et al., 2022) framework and adhered to the training and evaluation protocols outlined in Wei et al. (2024). The evaluation was performed on the *GTAV → CityScapes* setup (Richter et al., 2016; Cordts et al., 2016). As shown in Table 12, ProPETL significantly outperforms its counterparts, demonstrating its generalization ability under cross-scenario task.

Table 12: Domain generalization semantic segmentation results. $^*$: results from Wei et al. (2024).

| Method | CityScapes (mIoU) | Param. (M) |
|---|---|---|
| Full fine-tuning$^*$ | 63.7 | 304.20 |
| Freeze$^*$ | 63.3 | **0** |
| AdaptFormer$^*$ | 64.9 | 3.17 |
| Rein$^*$ | 66.4 | 2.99 |
| ProPETL | **67.9** | 3.17 |

**Intermediate Task.** In the main body of the paper, we present the ablation results of intermediate tasks from the perspectives of perception granularity and supervision diversity. To mitigate perturbations, we average results from experiments with varying supervision diversities at each perception granularity level, and vice versa. Here, we provide the full results of intermediate tasks comprising various combinations of perception granularity and supervision diversity on the VOC2012, COCO-Stuff10k, and CityScapes datasets. The results are illustrated in Table 13. As shown in this table, the combination of image-level perception and multi-label diversity yields the optimal performance, leading us to adopt this configuration within our framework.

**Visualizations.** To further investigate the impact of the proposed ProPETL framework, we visualize the comparisons of normalized Euclidean distance between the feature maps before and after adaptation in Fig. 8. As illustrated in the figure, representative methods such as BitFit and Adapt-Former tend to make localized adjustments, often overlooking broader perceptual regions. Conversely, LoRA generally implements globally consistent adjustments but may neglect challenging areas, such as edges. The feature adjustment pattern observed in VPT resembles that of full fine-tuning; however, it exhibits a messy adjustment, particularly in the last row of examples. ProPETL demonstrates an adaptation pattern akin to full fine-tuning, effectively capturing semantic adjustments with fine granularity while also focusing on critical edge regions. This highlights the efficacy

Table 13: Ablation study on the intermediate task using the DSA during downstream fine-tuning. "$s$" indicates the size of the pooling window. By adjusting $s$, the perception granularity of the intermediate tasks ranges from patch-level ($s = 32$) to image-level ($s = 640$). "$\alpha$" indicates the smoothing factor, respectively. By adjusting $\alpha$, the supervision diversity of intermediate tasks transitions from single label ($\alpha = 0$) to multi label ($\alpha = 1$). The first line is the result of AdaptFormer.

| Perception Granularity | | | Supervision Diversity | | | VOC2012 | | COCO-Stuff10k | | CityScapes | |
|---|---|---|---|---|---|---|---|---|---|---|---|
| $s = 32$ | $s = 128$ | $s = 640$ | $\alpha = 0$ | $\alpha = 0.5$ | $\alpha = 1$ | mIoU | mAcc | mIoU | mAcc | mIoU | mAcc |
| | | | | | | 85.41 | 90.93 | 45.45 | 57.83 | 79.41 | 86.03 |
| ✓ | | | ✓ | | | 85.72 | 91.27 | 45.74 | 57.90 | 80.73 | 86.94 |
| ✓ | | | | ✓ | | 85.63 | 90.96 | 45.50 | 57.19 | 80.87 | 87.24 |
| ✓ | | | | | ✓ | 85.91 | 91.67 | 45.57 | 58.31 | 80.92 | 87.61 |
| | ✓ | | ✓ | | | 85.53 | 91.78 | 45.76 | 58.15 | 80.85 | 87.26 |
| | ✓ | | | ✓ | | 85.54 | 91.96 | 46.11 | 57.96 | 80.89 | 87.56 |
| | ✓ | | | | ✓ | 85.79 | **92.39** | 46.09 | 58.00 | 81.35 | 87.92 |
| | | ✓ | ✓ | | | 85.70 | 91.71 | 46.09 | 58.20 | 80.09 | 86.99 |
| | | ✓ | | ✓ | | 85.87 | 91.26 | 46.40 | 58.11 | 80.91 | 87.39 |
| | | ✓ | | | ✓ | **86.11** | 92.08 | **46.48** | **58.69** | **81.67** | **88.23** |

of the proposed progressive paradigm. In Fig. 9, we visualize several images and corresponding segmentation results in the VOC2012 and ADE20k dataset. Compared to AdaptFormer, ProPETL produces smoother segmentation contours (first row) and more complete object masks (second and third rows). The last two rows illustrate failure cases where ProPETL struggles with segmentation redundancies in challenging scenarios, such as object occlusion (fifth row) and regions with high foreground-background similarity (last row).

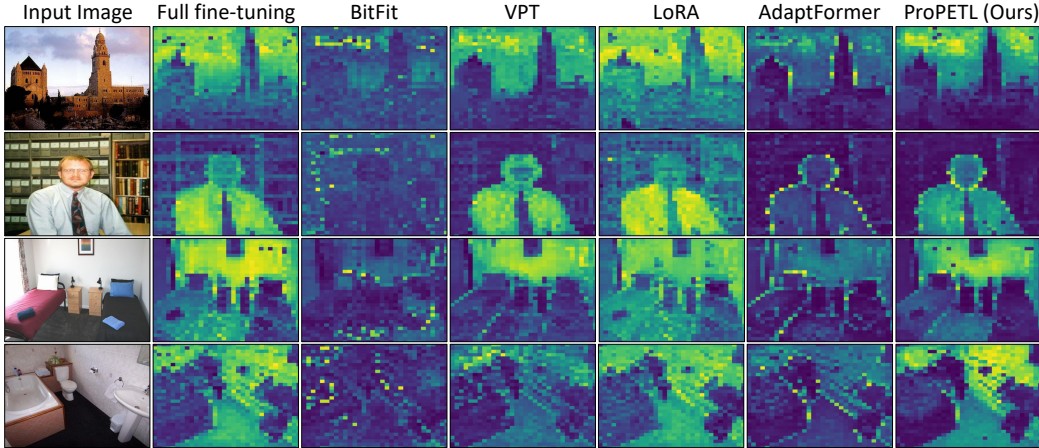

Figure 8: Visualization of the normalized Euclidean distance between the feature map before and after model adaptation on the ADE20k dataset.

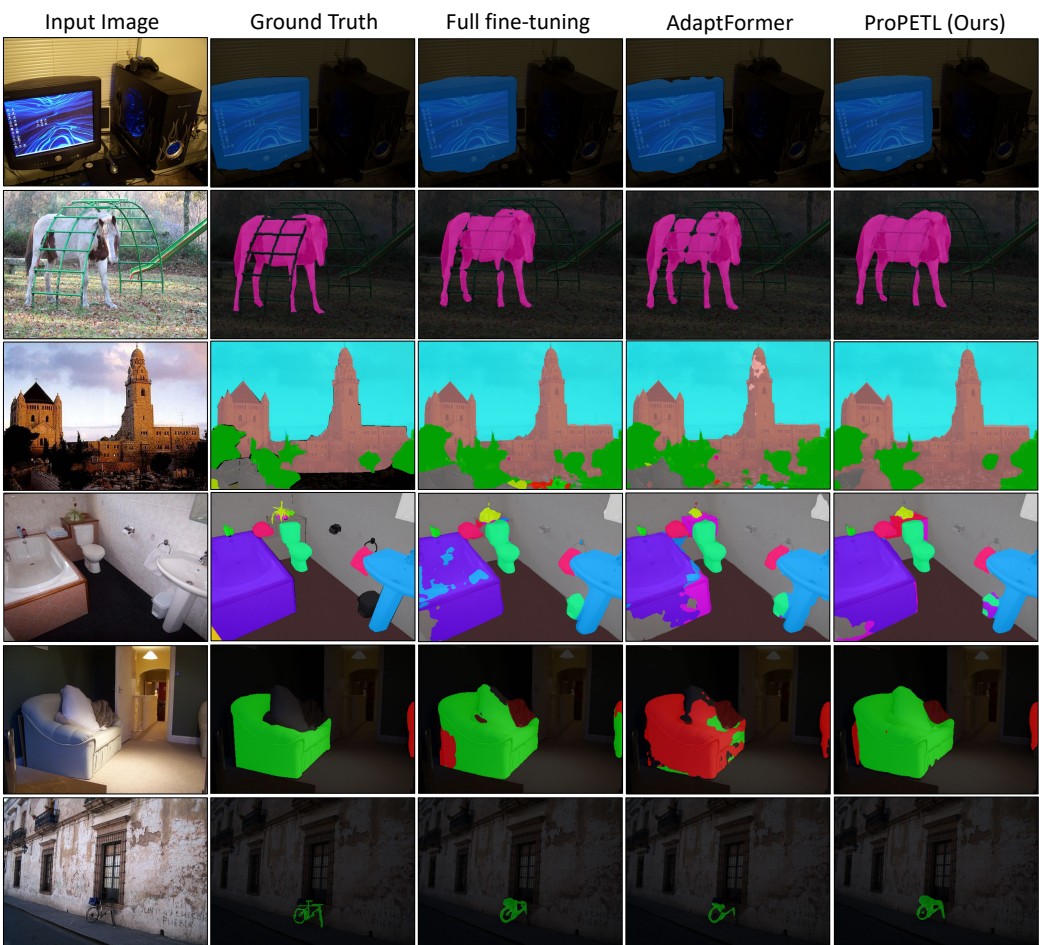

Figure 9: Visualization of the segmentation results on the VOC2012 and ADE20k dataset.

