# OpenReview forum: "Progressive Parameter Efficient Transfer Learning for Semantic Segmentation"
_ICLR.cc/2025/Conference — ICLR 2025 Poster_

### Official Review · Reviewer_Fur2 · 2024-10-27

**Soundness:** 3
**Presentation:** 3
**Contribution:** 3
**Rating:** 6
**Confidence:** 4

**Summary:**

This paper proposes a framework for progressively adapting pre-trained models to downstream semantic segmentation tasks through intermediate tasks, where perception granularity and supervision diversity are interesting and novel. Although this work has some weaknesses in descriptions and experiments, its overall contribution benefits the PETL community, especially in fine-tuning VFMs from classification tasks to semantic segmentation tasks. Thus, I recommend the weak acceptance of this paper and hope the authors' responses.

**Strengths:**

1. The motivations of the proposed approach are clear.
2. The intermediate task designation, including perception granularity and supervision diversity, is interesting and novel.
3. The proposed approach achieves better performance on multiple benchmarks.

**Weaknesses:**

1. The evaluations of the generalization ability are insufficient. ProPETL demonstrates strong generalization across various pre-trained models and scenarios. However, the experimental evaluation does not show the proposed ProPETL's generalization ability under cross-scenario tasks, such as the domain generalization setting. I am interested in the proposed ProPETL's generalization ability under cross-scenario tasks.
2. The descriptions of the proposed approach are unclear. For example, the summary of the overall training and inference process and the design of the GPA's architecture in intuition or theory.
3. Do α of supervision diversity and pooling window of perception granularity correspond one to one?
4. The computational cost of the proposed approach during the training and inference stages should be introduced and compared with other fine-tuning strategies in detail, such as the total training time, inference time, FLOPs, GPU Memory, Storage, etc.
5. More backbones, such as DINOv2 and EVA02, and more segmentation heads, such as Mask2Former and SemFPN, should be included to verify the effectiveness of the proposed approach.
6. How to evaluate and compare to the Rein [1] which also introduces the PETL method for semantic segmentation tasks.
[1] Stronger, Fewer, & Superior: Harnessing Vision Foundation Models for Domain Generalized Semantic Segmentation. CVPR 2024.
7. The fail examples, limitations, and future works should be provided and discussed.
8. Figure 1 (a) and (b) should be more elegant. Labels of the x-axis are too narrow.
9. The code should be released.

**Questions:**

See Weaknesses.

---

> ### Author Response · Authors · 2024-11-21
>
> **[R4-W1]:** Cross Scenerio Tasks
>
> **[R4-A1]:** As suggested, we conducted a comparison with Full Fine-tuning, AdaptFormer, and Rein under domain generalization settings. DINOv2 and Mask2Former were utilized as backbones, following the official implementations of Rein[a]. The comparison was performed under *GTAV → Cityscapes* setup. The results demonstrate that ProPETL outperforms all counterparts with comparable parameters, underscoring its strong generalization capability in cross-scenario tasks.
>
> We have included this analysis in the revised version to provide a more comprehensive evaluation of ProPETL's generalization ability.
>
> |                  | CityScapes (mIoU) | Param. (MB) |
> | ---------------- | :---------------: | :---------: |
> | Full fine-tuning |       63.7        |   304.20    |
> | AdaptFormer      |       64.9        |    3.17     |
> | Rein             |       66.4        |    2.99     |
> | ProPETL          |     **67.89**     |    3.17     |
>
> [a] Wei et al. Stronger, Fewer, & Superior: Harnessing Vision Foundation Models for Domain Generalized Semantic Segmentation. In CVPR 2024.
>
> **[R4-W2]:** Descriptions
>
> **[R4-A2]:**
>
> * **Overall Training and Inference Process**: The ProPETL training pipeline consists of two phases: midstream adaptation and downstream fine-tuning. In the midstream adaptation phase, we leverage image-level, multi-label intermediate tasks to train the adaptation module. During downstream fine-tuning, we adjust the adaptation module using the DSA strategy, followed by joint training with the segmentation head on the target segmentation task. For inference, the trained adaptation module, combined with the pre-trained backbone, extracts feature maps from input images, which are then fed to the segmentation head to generate the final segmentation results.
>
> * **Design of GPA's Architecture**: The adaptation module in GPA follows a bottleneck architecture consisting of a down-projection layer, a ReLU activation, and an up-projection layer. Individual adaptation modules are inserted into each transformer layer. The intuition behind the design is to allow GPA to pre-initialize adaptation module parameters through optimization on intermediate tasks before downstream fine-tuning.
>
> We will include more detailed descriptions in the final version and carefully proofread the manuscript to ensure clarity and precision.
>
>
>
> **[R4-W3]:** Corresponding of Supervision Diversity and Perception Granularity
>
> **[R4-A3]:** The parameter $\alpha$ (which controls supervision diversity) and the pooling window size (which controls perceptual granularity) are independently adjustable. There is no direct correspondence between them, as they govern different aspects of the intermediate task. We have revised the description in the experimental section to prevent any potential misunderstanding.
>
>
>
> **[R4-W4]** Computation Cost
>
> **[R4-A4]:** Please refer to **[R2-A2]** for a detailed comparison and analysis. We have included the computation cost comparison in the revised version.

---

> ### Author Response · Authors · 2024-11-21
>
> **[R4-W5]** Evaluation with More Backbones and Heads
>
> **[R4-A5]:** As suggested, we conducted additional comparisons in the DINOv2 + Mask2Former setting with the counterparts on VOC2012 and CityScapes. The results are presented below. From the results, we observe that ProPETL continues to outperform the counterparts in this setting. We have included results from additional settings in the revised version to more effectively demonstrate the superiority of ProPETL.
>
> |                  | VOC2012 (mIoU/mAcc) | CityScapes (mIoU/mAcc) | Params. (M) |
> | ---------------- | :-----------------: | :--------------------: | :---------: |
> | Full fine-tuning |     86.81/93.35     |      84.10/90.57       |   304.19    |
> | AdaptFormer      |     88.93/94.34     |      83.83/90.87       |    4.74     |
> | ProPETL          |   **89.53/94.79**   |      **84.29/91.37**      |    4.74     |
>
>
>
> **[R4-W6]:** Comparison to Rein
>
> **[R4-A6]:** We compared ProPETL with Rein under the following semantic segmentation settings:
>
> 1. **Cross-domain evaluation:** *GTAV (Train) → CityScapes (Test)*, as utilized in the Rein literature.
> 2. **In-domain evaluation:** *CityScapes (Train) → CityScapes (Test)*, as presented in our paper.
>
> In the cross-domain setting, ProPETL outperforms Rein (**67.89** vs. **66.4**, as noted in **[R4-A1]**). For the in-domain setting, since Rein did not report results, we conducted experiments using a supervised pre-trained ViT-Large backbone with a SETR [b] head. As shown in the table below, ProPETL achieves a 1.23% improvement in mIoU over Rein on CityScapes. These results demonstrate that ProPETL consistently outperforms Rein across both settings, underscoring its effectiveness.
>
> |             | CityScapes (mIoU/mAcc) | Param. (M) |
> | ----------- | ---------------------- | ---------- |
> | AdaptFormer | 75.42/83.99            | 3.17       |
> | Rein        | 75.61/83.34            | 2.99       |
> | ProPETL     | **76.84/84.96**        | 3.17       |
>
> [b] Zheng et al. Rethinking semantic segmentation from a sequence-to-sequence perspective with transformers. In CVPR 2021.
>
> **[R4-W7]:** Failed examples, limitations and future work
>
> **[R4-A7]:**
>
> * **Failure Cases:** We have revised the manuscript to include an analysis of failure cases. As illustrated in Figure 9 in the appendix, ProPETL encounters challenges with segmentation redundancy in scenarios involving object occlusion (fifth row) and high foreground-background similarity (last row). Addressing these issues may involve optimizing intermediate tasks based on the data distribution characteristics.
>
> * **Limitations and Future Work:** Looking ahead, several avenues merit further investigation. On one hand, while the midstream adaptation phase enhances fine-tuning performance, it does increase the overall computational cost of training. Exploring methods to shorten the global fine-tuning process and further improve fine-tuning efficiency remains an open challenge. On the other hand, the current design of intermediate tasks primarily relies on empirical studies. Developing dynamic approaches for designing intermediate tasks and generalizing ProPETL to accommodate a broader range of pretrained models and downstream scenarios are promising directions for future research.
>
>
>
> **[R4-W8]:** Figure 1.
>
> **[R4-A8]:**  We have refined Figures 1(a) and 1(b) in the revised manuscript for improved clarity. And we will try our best to proofread the manuscript in the final version.
>
>
>
> **[R4-W9]:** Code Release
>
> **[R4-A9]:** We will release the code and checkpoints to ensure reproducibility and will make every effort to assist the community in reproducing the results upon acceptance.

---

> > ### Comment · Reviewer_Fur2 · 2024-11-24
> > **Thanks for your reply**
> >
> > Thank you for the responses. I have decided to keep the original score of 6.

---

### Official Review · Reviewer_fzgQ · 2024-10-30

**Soundness:** 3
**Presentation:** 3
**Contribution:** 3
**Rating:** 6
**Confidence:** 3

**Summary:**

This paper presents ProPETL framework, which applies progressive adaptation to Parameter-Efficient Transfer Learning (PETL) and introduces a novel midstream adaptation stage to address perception granularity differences and supervision diversity in semantic segmentation tasks.

**Strengths:**

This paper analyzes two candidate progressive adaptation strategies: Generalized Parametric Adaptation (GPA) and Decoupled Structured Adaptation (DSA), and employs DSA to effectively mitigate the "forgetting" issue observed in GPA.

Extensive comparisons demonstrate that ProPETL delivers superior performance across benchmarks, underscoring its effectiveness and superiority.

**Weaknesses:**

1. Both Generalized Parametric Adaptation (GPA) and Decoupled Structured Adaptation (DSA) implement dimensionality reduction and expansion through bypass branches to reduce the number of parameters. However, the fixed reduction ratio $r$ may lead to the loss of critical feature information, especially in tasks with high perceptual demands (such as semantic segmentation). The fixed ratio of the reduction operation lacks adaptability to the importance of features and the complexity of input data, which may result in the bypass branch being unable to effectively retain fine-grained information, thus affecting the overall performance of the model.

2. This paper suggests that Perception Granularity and Supervision Diversity are key factors in the design of intermediate tasks, but it lacks theoretical support for these design choices. For example, it does not explain why image-level perception granularity and multi-label supervision can enhance the fine-grained performance of downstream tasks.

**Questions:**

1. DSA divides the adapter parameters into two independent parts: the parameters for the midstream adaptation stage( $\phi$) and the parameters for the downstream fine-tuning stage \($ \psi$ \). During the intermediate adaptation phase, DSA only optimizes $\phi$, while in the downstream fine-tuning phase, $\phi$ remains frozen, and only $ \psi$ is optimized. However, the strategy of freezing  $\phi$ during the downstream task, which reduces the parameter space that can be optimized and increases the burden on optimizing  $ \psi$  , leading the model to seek the best solution in a smaller space. For complex downstream tasks, this limitation may affect the model's  performance, as the optimization of  $ \psi$ cannot fully replace the tuning of  $\phi$.

---

> ### Author Response · Authors · 2024-11-21
>
> We sincerely appreciate Reviewer 3 (fzgQ) for the positive and insightful feedback on our work. Below, we provide detailed responses to address your concerns.
>
> **[R3-W1]:** Fixed Reduction Ratio
>
> **[R3-A1]:**  The primary goal of our work is to bridge the gap between pre-training and downstream tasks through a progressive adaptation process while maintaining a low parameter overhead. Thus, the role of $r$ is more to regulate the learnable parameters, mitigating the trivial effects of parameter fluctuations while achieving better trade-off between performance and parameter overhead. Regarding your concern that a fixed $r$ might result in the loss of critical feature information, we evaluate the performance under different values of $r$, and the results are presented below.
>
> | Reduction ratio | CityScapes (mIoU/mAcc) | Params. (MB) |
> | --------------- | ---------------------- | ------------ |
> | 48              | 81.67/88.23            | 2.64         |
> | 24              | 81.65/88.31            | 5.28         |
> | 12              | 81.29/87.87            | 10.56        |
>
> From the results, we observe that decreasing $r$ does not yield significant improvements in performance but does incur additional parameter overhead. This suggests that directly decreasing $r$ (add more parameters) may not address the bottleneck in the fine-tuning process for segmentation scenarios. Instead, the progressive paradigm appears to be more critical in this context. Therefore, we retain the original settings for their higher efficiency.
>
> **[R3-W2]:** Theoretical Support of Intermediate Task Designation.
>
> **[R3-A2]:** We appreciate the reviewer's insightful observation. Here, we would like to share some of our theoretical insights into the rationale behind our intermediate task design. Considering the standard formulation of the pixel-wise cross-entropy loss, $L_{\rm piexl-wise \ CE}$ for downstream segmentation tasks:
>
> $$
> L_{\rm piexl-wise \ CE}=-\frac{1}{HW}\sum_{i=1}^{HW}\sum_{c=1}^{C}y_{i,c}\log(\hat{p}_{i,c}),
> $$
>
> where $H$ and $W$ represent the height and width of the input, $C$ is the total number of classes, and $y_i$ is the one-hot encoded label vector for pixel $i$, with $y_{i,c} = 1$ if pixel $i$ belongs to class $c$. The term $\hat{p}_{i,c}$ denotes the predicted probability (post-softmax) for class $c$.
>
> The **perceptual granularity** in intermediate task design adjusts the perception window sizes, thereby altering the outer loop, which governs the global supervision intensity. On the other hand, changes in **supervision diversity** impact the inner loop, determining the number of classes to classify. By combining these two dimensions, we can effectively *interpolate* between image-level classification and pixel-level segmentation tasks. This creates a spectrum of intermediate tasks that bridge the gap between pretraining and downstream fine-tuning. We evaluated typical combinations of these factors in the main paper to assess the effectiveness of midstream tasks.
>
>
> **[R3-Q1]** Decoupled Structured Adaption
>
> **[R3-A3]:** As suggested, we conducted an additional quantitative comparison between simultaneous tuning and progressive tuning. The results are presented below. While simultaneous tuning offers larger optimization spaces, it is more challenging to achieve superior results due to the issue of knowledge forgetting, as illustrated in Figure 7 of the main paper.
>
>
> |                                               | VOC (mIoU/mAcc) |
> | --------------------------------------------- | :-------------: |
> | DSA (default: freezing $\phi$, tuning $\psi$) | **86.11/92.08** |
> | DSA (tuning both $\phi$ and $\psi$)           |   85.80/91.77   |
>
> Please feel free to let us know if you have any further questions.

---

> > ### Comment · Reviewer_fzgQ · 2024-11-22
> > **Thanks for the reply**
> >
> > I am of the opinion that this paper meets and exceeds the criteria for acceptance.

---

### Official Review · Reviewer_GSCi · 2024-11-01

**Soundness:** 3
**Presentation:** 3
**Contribution:** 2
**Rating:** 6
**Confidence:** 3

**Summary:**

The paper presents ProPETL, a new method to improve Parameter Efficient Transfer Learning (PETL) for semantic segmentation tasks, which usually have more difficulties in fine-tuning compared to classification tasks. Current PETL methods find it hard because they need to adjust more parameters to handle changes in the details of semantic understanding. ProPETL solves this problem by adding a midstream adaptation step, which gradually aligns the pre-trained models for segmentation tasks. This proposed method achieves the state-of-the-art performance, even better than full fine-tuning, on various evaluation benchmarks.

**Strengths:**

1. The authors provide a detailed analysis of why existing PETL methods do not perform effectively for semantic segmentation tasks. This paper not only introduces ProPETL but also thoroughly tests its effectiveness compared to state-of-the-art methods across multiple benchmark datasets. The authors conduct extensive ablation studies to show the effect of each component of the method, which supports the strength of the proposed contributions.

2. The paper is well-written and organized clearly, making the paper easy to understand. The authors include background information and visualize to illustrate the weaknesses of current PETL methods and the strength of ProPETL.

3. ProPETL outperforms full fine-tuning on various benchmarks, showing that parameter-efficient methods can match or even exceed traditional methods.

**Weaknesses:**

1. The proposed midstream adaptation phase makes the whole framework more complex than previous methods. Selecting intermediate tasks, such as perception granularity and supervision diversity, looks like it’s based on trial and error without a clear, systematic way for users to apply these choices to new datasets or tasks.

2. While the progressive adaptation process improves segmentation performance, it may need more computational time because of the extra midstream phase, especially for large datasets. The paper claims that this extra cost is worth it for better performance, but having more exact comparisons of training time or resource usage with other PETL methods would help show the practical trade-offs, especially when computational efficiency is very important.

**Questions:**

1. The paper shows that ProPETL works well with a Swin-L model pre-trained on ImageNet-21K, but how would the performance change if we use smaller or less powerful pre-trained models? Is there any guideline or expectation for using ProPETL in situations with limited resources? Would ProPETL still be helpful in transfer learning cases that use self-supervised models with lower representational capacity? It would be good to see more discussion or results about how much ProPETL depends on the quality of pre-trained models.

2. The paper says that the midstream adaptation improves segmentation performance, but how does the computational cost compare with traditional PETL methods, in terms of training time?

---

> ### Author Response · Authors · 2024-11-21
>
> We sincerely appreciate Reviewer 2 (GSCi) for the positive and insightful feedback on our work. Below, we provide detailed responses to address your concerns.
>
>
>
> **[R2-W1]** Complexity and Generalization.
>
> **[R2-A1]:** The proposed ProPETL introduces an additional phase into the downstream fine-tuning process without affecting inference efficiency, which aligns with the more common application scenario. A detailed comparison is provided in **[R2-A2]**. Additionally, the global computational cost is comparable to state-of-the-art methods, while achieving significantly superior performance on downstream applications.
>
> In this paper, we aim to thoroughly investigate the degradation of fine-tuning performance in segmentation tasks. Our key insight is that semantic perceptual granularity shifts necessitate larger adjustments. It is worth noting that our primary objective is not to determine the optimal combination of perceptual granularity and supervision diversity, but rather to highlight that introducing midstream adaptation effectively bridges this gap and improves downstream performance. Consequently, we evaluate typical configurations of these two factors. In our experiments, we observed several trends in intermediate task design. Greater supervision diversity ($\alpha=1$) consistently leads to improved performance across all settings, suggesting that diverse supervision in the midstream adaptation phase better equips the model for downstream fine-grained dense predictions, as required in segmentation scenarios. Regarding perceptual granularity, performance varies across datasets, potentially influenced by the inductive biases of the downstream datasets. However, larger perceptual granularity generally delivers better results compared to patch-level settings ($s=32$).
>
>
>
> **[R2-W2]:** Computation cost.
>
> **[R2-A2]:** As suggested, we compare the proposed ProPETL with its counterparts in terms of computational cost. The results are presented below.
>
>
>
> |                  | Training time (H) | Inference time (Ms) | Workload (GFLOPs) | Peak GPU memory (MB) | COCO-Stuff10k (mIoU) |
> | ---------------- | :---------------: | :-----------------: | :---------------: | :------------------: | :------------------: |
> | Full fine-tuning |        41         |       **182**       |    **623.310**    |        94,198        |        46.30         |
> | VPT              |        96         |         210         |      872.222      |       135,498        |        45.04         |
> | LoRA             |        30         |         208         |      718.097      |        94,564        |        45.03         |
> | AdaptFormer      |        34         |         185         |      627.085      |      **51,433**      |        45.45         |
> | ProPETL          |        47         |         191         |      627.111      |        55,613        |      **46.48**       |
> | ProPETL (light)  |      **29**       |         191         |      627.111      |        55,613        |        45.98         |
>
> From the results, we observe that the inference time, workload, and peak GPU memory usage of ProPETL are comparable to state-of-the-art methods. Its only drawback is a slightly higher training time due to the additional phases. However, this training process is performed only once per fine-tuning session and delivers significantly superior performance in downstream applications. To further enhance the practicality of ProPETL, we introduce a lightweight version that reduces the duration of the second phase (i.e., downstream fine-tuning) by half. This adjustment results in a lower overall training time compared to all counterparts while still achieving significantly better performance in downstream tasks, underscoring the superiority of our approach.
>
> We have included this comparison in the revised version to provide a more comprehensive evaluation of the prodposed ProPETL.

---

> ### Author Response · Authors · 2024-11-21
>
> **[R2-Q1]:** Performance with Smaller/Less Powerful Pretrained Models
>
> **[R2-A3]:** As suggested, we validated ProPETL using backbones with varying sizes and pretrained strategy. The results are presented below.
>
> For the evaluation of model size, we compared Swin-Transformers of various scales, ranging from small to large, on the Pascal VOC 2012 dataset. The results indicate that ProPETL consistently outperforms the baseline across all settings. Furthermore, as the model size increases, the benefits of the midstream adaptation process become more pronounced, resulting in greater performance improvements.
>
> |                  | Swin-small      | Swin-base       | Swin-large      |
> | ---------------- | --------------- | --------------- | --------------- |
> | Full fine-tuning | 80.83/86.90     | 82.07/87.63     | 84.38/89.82     |
> | AdaptFormer      | 80.94/87.26     | 84.29/90.31     | 85.41/90.93     |
> | ProPETL          | **81.26/87.82** | **85.03/90.96** | **86.11/92.08** |
>
> For the evaluation of model capacity, we have assessed the generalization of ProPETL on models pretrained using MAE, as shown in Table 4 of the main paper. To further evaluate generalization, we employed the ViT-Base pretrained with MoCo-v3 as the backbone and conducted comparisons with full fine-tuning and state-of-the-art AdaptFormer. The results demonstrate that ProPETL significantly outperforms the baseline and linear probing by a large margin. Combining these findings with the evaluations in Tables 1 and 4 of the main paper, we conclude that ProPETL consistently delivers improvements across pretrained backbones of varying scales and capacities.
>
> |                  | CityScapes (mIoU/mAcc) | Params. (MB) |
> | ---------------- | ---------------------- | ------------ |
> | Full fine-tuning | 64.58/73.26            | 85.84        |
> | Linear Probing   | 42.02/48.47            | 0            |
> | AdaptFormer      | 54.93/62.86            | 1.19         |
> | ProPETL          | **58.83/67.00**        | 1.19         |
>
> We have included these results in the revised version to provide a more comprehensive evaluation of ProPETL.
>
> **[R2-Q2]:** Computation Cost and Training Time
>
> **[R2-A4]:** Please kindly refer to **[R2-A2]** for a detailed comparsion and analysis.
>
>
>
> Please feel free to let us know if you have any further questions or require additional clarifications.

---

> > ### Comment · Reviewer_GSCi · 2024-11-22
> >
> > Thank you for the comment. The reviewer is satisfied with the response and has decided to keep the original score of 6

---

### Official Review · Reviewer_4Nue · 2024-11-03

**Soundness:** 3
**Presentation:** 3
**Contribution:** 3
**Rating:** 6
**Confidence:** 4

**Summary:**

The paper finds that there is a perceptual granularity gap between pre-trained models and downstream segmentation tasks, necessitating significant parameter adjustments. However, existing PETL approaches fail to effectively address this gap due to the constraint of limited trainable parameters. To tackle this issue, the paper introduces ProPETL, which incorporates an additional midstream adaptation phase to progressively align pre-trained models with downstream segmentation tasks. Specifically, the paper conducts an in-depth analysis of candidate progressive adaptation strategies, such as Generalized Parametric Adaptation and Decoupled Structured Adaptation, as well as intermediate tasks designed to enhance the perception of finer granularity and diverse classes. Notably, ProPETL achieves state-of-the-art performance on most segmentation benchmarks, surpassing even full fine-tuning.

**Strengths:**

1. The paper addresses the gap between pre-trained models and downstream segmentation tasks by introducing an additional midstream adaptation process, while maintaining parameter efficiency.
2. The method demonstrates its superiority by outperforming other parameter-efficient approaches and even surpassing full fine-tuning.

**Weaknesses:**

1. The paper is motivated by the observation that the perceptual granularity gap between pre-trained models and downstream segmentation tasks necessitates significant parameter adjustments. However, for models like SAM[1], which are pre-trained on segmentation tasks, there is a lack of evidence to support that these observations still hold.
2. The necessity of additional midstream adaptation is not entirely clear, as the proposed intermediate tasks resemble data augmentation, which could potentially be integrated into the fine-tuning phase.

[1] Kirillov A, Mintun E, Ravi N, et al. Segment anything[C]//Proceedings of the IEEE/CVF International Conference on Computer Vision. 2023: 4015-4026

**Questions:**

Could the intermediate tasks be considered a form of data augmentation? Would performance improve if these intermediate tasks were integrated into the fine-tuning phase? For example, scaling the images to different sizes and generating varying numbers of categories at random ratios during fine-tuning.

---

> ### Author Response · Authors · 2024-11-21
>
> We sincerely appreciate Reviewer 1 (4Nue) for the positive and insightful feedback on our work. Below, we provide detailed responses to address your concerns.
>
> **[R1-W1]:** Observations for SAM.
>
> **[R1-A1]:** As suggested by R1, we selected two representative downstream tasks—classification (CIFAR-100) and segmentation (Cityscapes)—to examine the fine-tuning process on the segmentation-based pretrained model. Specifically, we calculated the Euclidean distance for SAM before and after fine-tuning. The results are presented below.
>
> |       | [CLS] Cifar100 | [SEG] CityScapes |
> | :---: | :------------: | :--------------: |
> | Block |     37.00      |      32.43       |
> |  MSA  |     24.35      |      18.81       |
> |  FFN  |     27.82      |      25.85       |
>
> From the results, we observe that the effect of the granularity gap on parameter adjustment remains valid in this scenario. When fine-tuning the segmentation-pretrained SAM for downstream segmentation tasks (CityScapes), the magnitude of parameter adjustment is significantly smaller compared to fine-tuning for the classification task. This indicates that the perceptual granularity gap necessitates larger parameter adjustments. These observations further support our insights and complement the findings of the previous experiments.
>
> To furtherly evaluate the proposed ProPETL in this scenerio, the results are illustrated as bellow.
>
> |             | CityScapes (mIoU/mAcc) | Params. (M) |
> | ----------- | :--------------------: | :---------: |
> | AdaptFormer |      79.02/88.05       |    6.32     |
> | ProPETL     |    **80.24/89.10**     |    6.32     |
>
> From this comparison, we observe that the progressive adaptation process also enhances the fine-tuning performance for segmentation tasks, regardless of whether the backbone is pretrained for classification or segmentation. This further demonstrates the effectiveness of the proposed approach. We will include these additional comparisons in the final version to provide a more comprehensive evaluation of the proposed methods.
>
>
> **[R1-W2-Q1]:** The Necessity of Midstream Adaptation, Its Relationship with Data Augmentation, and Its Integration with Fine-Tuning
>
> **[R1-A2]:** Building on the observation that a larger perceptual granularity gap between pretrained models and downstream tasks necessitates more substantial adjustments, introducing a midstream step to bridge this gap during the fine-tuning process emerges as a natural solution. This paradigm has proven effective in applications such as [a] and [b]. However, this progressive approach remains largely unexplored in the field of PETL. To address this limitation, we conducted a detailed analysis of midstream task designations and progressive strategies in this work. The experiments presented in Tables 2 and 3, as well as Figure 6 in the main paper, validate the effectiveness of midstream adaptation.
>
> We respectfully differentiate the proposed midstream adaptation from classical data augmentation, as they address different aspects of the training process. Data augmentation perturbs the input data, encouraging the model to produce consistent predictions. In contrast, midstream adaptation retains the same inputs but modifies the annotations by incorporating different granularities and diverse forms of supervision. To further compare the proposed midstream adaptation with data augmentations and evaluate its integration with fine-tuning, we employ AdaptFormer as the baseline and conduct segmentation fine-tuning on the VOC2012 dataset. The results are presented below.
>
> |                                | VOC2012 (mIoU/mAcc) |
> | ------------------------------ | :-----------------: |
> | AdaptFormer                    |     85.41/90.93     |
> | AdaptFormer (w/data aug.)      |     85.40/91.25     |
> | AdaptFormer (w/joint training) |     85.65/91.39     |
> | ProPETL                        |   **86.11/92.08**   |
>
> The results indicate that while data augmentations slightly improve mAcc, they lead to a drop in mIoU, as they fail to address the granularity gap due to their sole focus on modifying the input images. In contrast, when ProPETL is employed, the performance significantly improves to 86.11 mIoU and 92.08 mAcc, highlighting the differences between midstream adaptation and data augmentation. Furthermore, integrating midstream adaptation with the fine-tuning process yields better performance than the baseline but still falls short of the default ProPETL. This discrepancy arises because midstream and downstream tasks emphasize different perceptual granularities, potentially causing conflicts that hinder model convergence and result in performance degradation.
>
> [a]  Weinshall et al. Curriculum learning by transfer learning: Theory and experiments with deep networks. In ICML 2018.
>
> [b] Hsu et al. Progressive domain adaptation for object detection. In WACV 2020.
>
> Please feel free to let us know if you have any further questions or require additional clarifications.

---

> > ### Comment · Reviewer_4Nue · 2024-11-24
> > **Official Comment by Reviewer 4Nue**
> >
> > Thank you for the detailed response. I will raise the score to 6.

---

### Author Response · Authors · 2024-11-21
**Common response**

We sincerely appreciate the detailed and constructive feedback from all the reviewers. In particular, we are grateful for their recognition of the work's contribution to the PETL community (Fur2) and their acknowledgment of the idea as novel (fzgQ, Fur2), interesting (Fur2), and supported by clear motivation (Fur2) and strong writing (GSCi). Moreover, we are pleased that the method has been recognized as well-analyzed (4Nue, GSCi) and well-performed (4Nue, GSCi, fzgQ, Fur2).

Before addressing the individual reviews, we briefly outline the manuscript changes and key points highlighted in the feedback:
1. **Additional Analysis**:
   - Added analysis on computational overhead.
   - Updated sections on failure cases, limitations, and future work.
2. **Extended Evaluation**:
   - **Pre-trained Models**: Additional evaluations using various sizes of Swin-Transformers, MoCo-v3, SAM, and DINOv2 as encoders.
   - **Segmentation Heads**: Mask2Former has been added as a segmentation head for evaluation.
   - **Cross-Domain Evaluation**: The effectiveness of ProPETL has been assessed under the *GTAV → CityScapes* domain generalization setting.
3. **Minor Edits**:
   - Refined descriptions of the training and inference processes.
   - Optimized Figure 1 for improved clarity.

---

### Meta-Review · Area_Chair_uBrv · 2024-12-23

**Metareview:**

This paper introduces a framework named ProPETL, which can adapt pre-trained models to downstream semantic segmentation tasks through intermediate tasks progressively, together with a new midstream adaptation stage to address perception granularity differences and supervision diversity. The manuscript was reviewed by four experts in the field. The recommendations are (4 x "6: marginally above the acceptance threshold"). Based on the reviewers' feedback, the decision is to recommend the acceptance of the paper. The reviewers did raise some valuable concerns like framework and computational complexity, unconvincing experimental evaluations, unclear descriptions and statements that should be addressed in the final camera-ready version of the paper. The authors are encouraged to make the necessary changes to the best of their ability.

**Additional Comments On Reviewer Discussion:**

Reviewers mainly hold concerns regarding framework and computational complexity (Reviewer GSCi), unconvincing experimental evaluations (Reviewer GSCi, fzgQ, Fur2), unclear descriptions and statements (Reviewer 4Nue, fzgQ, Fur2), and together with further polishment of the manuscript (Reviewer Fur2). The authors address these concerns with detailed and extra experiments and commit to polishing the revised version further.

---

### Decision · Program_Chairs · 2025-01-22

Accept (Poster)